# The Gut Microbiome in Depression and Potential Benefit of Prebiotics, Probiotics and Synbiotics: A Systematic Review of Clinical Trials and Observational Studies

**DOI:** 10.3390/ijms23094494

**Published:** 2022-04-19

**Authors:** Sauliha R. Alli, Ilona Gorbovskaya, Jonathan C. W. Liu, Nathan J. Kolla, Lisa Brown, Daniel J. Müller

**Affiliations:** 1Campbell Family Mental Health Research Institute, Centre for Addiction and Mental Health, Toronto, ON M5T 1R8, Canada; sauliha.alli@mail.utoronto.ca (S.R.A.); ilona.gorbovskaya@camh.ca (I.G.); jonathan.liu@camh.ca (J.C.W.L.); nathan.kolla@camh.ca (N.J.K.); 2Temerty Faculty of Medicine, University of Toronto, Toronto, ON M5S 1A8, Canada; 3Institute of Medical Sciences, University of Toronto, Toronto, ON M5S 1A8, Canada; 4Department of Pharmacology and Toxicology, University of Toronto, Toronto, ON M5S 1A8, Canada; 5Department of Psychiatry, University of Toronto, Toronto, ON M5T 1R8, Canada; 6Great Scott Consulting, New York, NY, USA; greatscottconsultingllc@gmail.com

**Keywords:** gastrointestinal microbiome, gut microbiota, gut–brain axis, major depressive disorder, depression, depressive symptoms, prebiotics, probiotics, synbiotics, systematic review

## Abstract

An emerging body of literature demonstrates differences in the gut microbiome (GMB) of patients with major depressive disorder (MDD) compared to healthy controls (HC), as well as the potential benefits of prebiotic, probiotic, and synbiotic treatment. We conducted a systematic review of 24 observational studies (n = 2817), and 19 interventional trials (n = 1119). We assessed alpha diversity, beta diversity, and taxa abundance changes in patients with MDD relative to HC, as well as the effect of prebiotics, probiotics, and synbiotics on depressive symptoms in individuals with clinical or subclinical depression. We observed no significant differences in alpha diversity but a significant difference in beta diversity between patients with MDD and HC. There were fluctuations in the abundance of specific taxa in patients with MDD relative to HC. Probiotic and synbiotic, but not prebiotic, treatment showed a modest benefit in reducing depressive symptoms in patients with MDD over four to nine weeks. The GMB profiles of patients with MDD differ significantly from HC, but further studies are needed to elucidate the benefits of prebiotic, probiotic and synbiotic treatments relative to antidepressants and over longer follow-up before these therapies are implemented into clinical practice.

## 1. Introduction

Major depressive disorder (MDD) is a mood disorder characterized by discrete episodes of at least two weeks involving changes in affect, cognition, and neurovegetative functions, with inter-episode remissions [1]. With an estimated lifetime prevalence of 10.8% in community samples globally [2], it is the most common psychiatric illness worldwide, and a leading cause of disability [3]. 

One of the most widely accepted models of depression pathophysiology has been the monoamine hypothesis, which postulates that the underlying pathophysiology of depression is a depletion of the neurotransmitters serotonin, norepinephrine, and dopamine in the central nervous system [4]. However, there are several limitations to this model, including the fact that up to 30% of patients with MDD do not respond to monoaminergic antidepressants [5], these medications are effective in psychiatric disorders with differing pathophysiologies [6], and disruptions of monoamine systems do not exacerbate existing depressive symptoms [7]. Therefore, there likely exist other mechanisms by which depressive pathologies arise which might open new avenues for treatment.

### 1.1. The Gut–Brain Axis in Depression

The gut–brain axis has been postulated to be involved in the onset of depression. The gut–brain axis is a bidirectional communication network between the gut and the brain that operates by neuroimmune and neuroendocrine processes [8,9,10,11]. It is mediated by several molecules, including short chain fatty acids [12], secondary bile acids, GABA neurotransmitters [13], and tryptophan metabolites, which are derived from the microbiota [14,15,16]. 

During dysbiosis, or a disruption to microbiota homeostasis, gut–brain pathways are dysregulated and associated with neuroinflammation and altered permeability of the blood–brain barrier [17]. Microbiota alterations may produce changes in depression by directly affecting release of the neurotransmitters serotonin and dopamine, influencing the stress response and hypothalamus–pituitary–adrenal (HPA) axis, influencing levels of brain-derived neurotrophic factor (BDNF) and triggering the release of inflammatory cytokines [18]. For example, depression is associated with the release of C-reactive protein (CRP) and cytokines such as IL-1, IL-2, IL-6, IFN-γ, and IL-1β [19]. For an in-depth review of the gut brain axis, see this 2019 review by Cryan and colleagues [20].

A study by Guida and colleagues (2017) revealed that antibiotic-induced dysbiosis in mice led to a general inflammatory state, and depressive-type behaviour, but was reversed with the probiotic *Lactobacillus casei* [21]. Fecal microbial transplant from humans with MDD to microbiota-deficient rodents has also been shown to induce a depression-like phenotype [22]. While human interventional trials are only beginning to gain traction, preliminary evidence from observational studies has shown that the gut microbiome (GMB) profiles of patients with MDD and depressive symptoms differ significantly from healthy controls (HC) [23,24,25,26,27], as well as patients with other mood [28,29] and anxiety disorders [30].

### 1.2. Prebiotic, Probiotic, Synbiotic, and Microbiota Therapeutics in Depression

Given this emerging evidence implicating the gut–brain axis in depression, there has been interest in developing treatments, namely probiotics, prebiotics, synbiotics, and microbiota restoration therapies that target the GMB. Probiotics are preparations of microorganisms that, when administered, improve gut microbial balance [31], while prebiotics are nondigestible compounds (e.g., fructooligosaccharides, galacto-oligosaccharides, and xylooligosaccharides) that are metabolized by gut microorganisms, modulating GMB composition to ultimately benefit the host [32]. The scientific literature classifies prebiotics as functional foods, given their role in promoting health and preventing disease [33]. When combined synergistically, probiotics and prebiotics are known as synbiotics [34]. Newer treatment modalities such as microbiota therapeutics, have yet to be evaluated in MDD but based on proof-of-concept hold promise for the treatment of MDD. Microbiota therapeutics include whole fecal microbiota transplants (FMT), symbiotic microbial consortia, or engineered symbiotic microbes [34]. The goal of microbiota therapeutics is to reconstitute a dysbiotic microbiota with a healthy microbiota.

Probiotics most often consist of combinations of *Lactobacillus* and *Bifidobacteria* genera. These microorganisms have been shown to suppress inflammation and modulate the immune system by preventing the induction of the cytokine IL-8 in human colon epithelium [35], as well as reduce intestinal permeability, inhibiting endotoxemia [36]. Several studies suggest that probiotic use confers physical and mental health benefits to the host [37,38,39], including as treatments for depression. In preclinical studies, administration of the probiotic *Bifidobacterium infantis* to rats has been shown to reverse experimentally-induced stress and depression [40], while supplementation with *Lactobacillus rhamnosus* for 28 days results in a decline in depressive symptom ratings [35]. A more recent study by Li and colleagues (2018) showed that in a chronic mild stress mouse model of depression and anxiety, there was a reduction in *Lactobacillus* species, and an increase in the inflammatory markers IFN-γ, TNF-α, and indoleamine 2,3-dioxygenase-1 levels in the hippocampus. Probiotic supplementation attenuated anxiety and depressive-like behaviors, significantly increased *Lactobacillus* abundance, and reversed immune changes [41]. These studies provide evidence that the antidepressant-like activity of probiotics may operate through a gut microbiota–inflammation–brain axis.

### 1.3. Previous Literature

Several reviews have also demonstrated a relationship between the GMB and major depressive disorder in human participants [26,42,43,44,45,46,47,48,49,50]. However, these studies tend to be restricted to patients who meet strict criteria for MDD, at the expense of including more common, subclinical forms of depression that are prevalent in inpatient and healthy populations [51,52] and may be more likely to respond to prebiotic, probiotic, and synbiotic supplementation [53]. In a recent systematic review and meta-analysis, Hofmeister and colleagues (2021) did report a statistically significant benefit of probiotic, prebiotic, and synbiotic interventions in people experiencing depressive symptoms (irrespective of MDD diagnosis) [48]. It is important to note, however, that they included patients with comorbid medical and psychiatric disorders who may have differing gut microbiome compositions than patients with depression and HC, and did not review evidence from observational studies [48].

The most recent systematic review of both observational studies and interventional trials was conducted by Sanada and colleagues in 2020 which included studies published until October 2019 [43]. The review synthesized evidence from ten observational studies that investigated differences in GMB diversity and taxa abundance in patients with MDD compared to HC, and six clinical trials that investigated changes in depressive symptom severity following probiotic or synbiotic administration. The authors report an overall effect of prebiotic and probiotic treatment on depressive symptoms, but inconsistent findings on GMB differences between MDD patients and healthy controls at the phylum level. Since then, an additional 13 interventional trials and 16 observational studies have been conducted which provide new evidence for our analysis. Thus, the present systematic review seeks to understand how the microbiota composition of patients with MDD or depressive symptoms differs from healthy controls, and the potential effects of prebiotic, probiotic and/or synbiotic treatment on depressive symptoms using an updated body of literature.

## 2. Results

### 2.1. Search Results

We report the process used to select the observational studies and clinical trials in two PRISMA Flow Diagrams [54].

In the search for observational studies (Figure 1), we identified 333 records through database searching. After removing 111 duplicates, we screened the title and abstracts of 222 articles, and assessed 37 full texts for eligibility. We also identified and screened three records from other reviews and websites. Ultimately, 24 observational studies (2817 participants) were included in our analysis.

In the search for clinical trials (Figure 2), we identified 382 records through database searching. After removing 71 duplicates, we screened the title and abstracts of 311 articles, and 26 full texts based on our eligibility criteria. We also identified and screened three records from other reviews and websites. We ultimately selected 19 observational studies (1119 participants) for inclusion in our analysis.

### 2.2. Findings from the Observational Studies

Characteristics of the 24 observational studies included in this analysis [22,23,24,25,26,27,28,29,30,55,56,57,58,59,60,61,62,63,64,65,66,67,68,69] are reported in Table 1. The sample consisted mainly of patients with MDD, who were predominantly females, with ages ranging from 20 to 80 years. Diagnoses were confirmed by the Diagnostic and Statistical Manual of Mental Disorders (DSM), Mini International Neuropsychiatric Interview (MINI), Structured Clinical Interview for DSM-5 (SCID) (version IV or V) or International Classification of Diseases-10 (ICD-10). We also included six studies that involved patients with IBS [56], BD [28,29,69], and anxiety [30,65] as they had subsamples of patients with MDD and HC alone. The majority of studies (19 out of 24) were recruited in Asia, but three studies were performed in North America, and two in Europe.

All studies conducted genetic analyses on the fecal microbiota (Table 2). Most studies conducted 16S rRNA gene sequencing, examining the V3-V5 regions of the genome, however Rhee and colleagues (2020) as well as Caso and colleagues (2021) used 16s rDNA [29,67]. Three studies reported shotgun metagenomic sequencing (SMS) [26,28,68] with one study reporting metaproteomic analysis [61]. The majority of studies reported no significant differences in the alpha diversity of patients with MDD and healthy controls (14 of 21). However, two thirds of the studies (12 of 18) showed that beta diversity significantly differed between patients with MDD and HC.

The abundance of bacterial taxa in patients with MDD relative to HCs, based on the results of 23 studies, are reported in Appendix A, and summarized in Table 3. The following taxa were increased in patients with MDD: the families Bifidobacteriaceae and Streptococcaceae (reported in four studies) as well as the genera Eggerthella (six studies) and Streptococcus (five studies). Conversely, there was a decrease in the phylum Bacteroidetes (four studies), family Sutterellaceae (four studies), genus Coprococcus (six studies), and genus Faecalibacterium (seven studies). We observed other changes in the relative abundance of bacterial species in three or fewer studies; however, the results are not described here.

### 2.3. Findings from the Clinical Trials

Characteristics of the 19 clinical trials included for synthesis [53,70,71,72,73,74,75,76,77,78,79,80,81,82,83,84,85,86,87] are reported in Table 4, and major findings from these studies in Table 5. The sample consisted of patients with MDD, a depressive episode, or depressive symptoms, with more females than males. The mean age of participants ranged from 20 to 40 years. Diagnoses were ascertained by DSM-IV, DSM-V, and ICD-10. Five studies included subclinical forms of depression defined by various scales, including the Quick Inventory of Depressive Symptomatology (QIDS-SR16) [71], the depression subscale of the Depression, Anxiety, and Stress Scale (DASS-42) [71], the Hamilton Depression Rating Scale (HAM-D) [70], and the Edinburgh Postnatal Depression Scale (EPDS) [83]. Studies involved populations from heterogeneous regions, including North America, Asia, Europe, and Australia.

About half of the studies were double-blind, randomized controlled trials (RCT) (11 of 19 studies), but results were also published in open label trials (four studies), pilot trials (two studies) and triple blind RCTs (one study). Most often, the intervention used was a probiotic (17 studies), followed by prebiotic (three studies) and synbiotics (one study). Probiotics mainly consisted of a combination of *Lactobacillus* and *Bifidobacterium* species. One prebiotic study explored the benefits of Inulin 10 g per day [78] relative to placebo. An additional two studies [75,79] compared the benefits of the prebiotic galactooligosaccharide relative to the probiotic CEREBIOME^®^ and placebo. Only one study [73] compared the benefits of a synbiotic (Familact H, a combination of fructooligosaccharide and the species *L. casaei*, *L. acidofilus*, *L. bulgarigus*, *L. rhamnosus*, *B. breve*, *B. longum*, and *S. thermophilus*) relative to placebo.

The majority of the trials were placebo-controlled (15 of 19), although some studies had no controls or used patients with antidepressant medication only as controls. The follow-up period for most studies was eight weeks, but ranged from four to nine weeks. Depressive symptoms were most often assessed by the Beck Depression Inventory (BDI) and HAM-D.

More than half (10 out of 17) of the probiotic studies demonstrated a significant decrease in the depressive symptoms of patients treated with probiotics over time, while six reported no significant decrease. In addition, one study reported mixed results, where there was a significant decrease in MADRS score in the probiotic group between baseline and four weeks, but no significant decrease between four and eight weeks [53]. Of the three studies examining the benefits of prebiotic treatment, none reported significant decreases in depressive symptom scores over an eight-week follow-up period [75,78,79]. However, significant decreases in symptoms were observed following synbiotic treatment for eight weeks in the single synbiotic study by Ghorbani and colleagues [73].

Five of the interventional trials also included microbiota analysis [27,77,80,85,86]. The majority of these studies (four out of five) reported no significant differences in alpha and beta diversity following probiotic administration. Results from four studies conducting taxa abundance analyses [80,85,86] revealed an increase in the proportion of *Ruminococcus gauvreauii*, *Coprococcus 3*, *Desulfovibrio*, *Faecalibacterium*, *Bifidobacterium*, *Adlercreutzia*, *Megasphaera*, and *Veillonella*, as well as a decrease in *Rikenellaceae_RC9_gut_group*, *Sutterella*, and *Oscillibacter* in patients treated with probiotics compared to healthy controls.

## 3. Discussion

### 3.1. Observational Studies

Here we examined the gut microbiota in patients with MDD based on 24 observational studies and 19 interventional trials. Taken together, the observational studies demonstrated no significant differences in alpha diversity in patients with MDD compared to HC. As alpha diversity is a measure of species richness and evenness within a single population [88], these findings suggest that the diversity of the GMB is similar for patients with MDD and HC.

Our findings are consistent with those of a recent systematic review of gut microbiome composition in patients with MDD, BD, and SZ compared to HC conducted by McGuinness and colleagues (2022) [89]. The authors observed no strong evidence for a difference in the alpha diversity of bacteria in patients with psychiatric disorders compared to HC [89]. Previously, a decrease in alpha diversity was hypothesized to exist in patients with psychiatric disorders. This was in line with the assumption that greater species number and diversity contributed to metabolic functional redundancy and resistance to pathogenic colonization [90,91], preventing disease. More recently, however, human gut microbiome studies suggest that there is limited utility of alpha diversity metrics in measuring gut health and distinguishing disease cases and controls. The evidence for alpha diversity changes in patients with MDD relative to HC is in fact largely mixed. For example, while McGuinness and colleagues (2022) reported no significant differences in alpha diversity between patients with psychiatric illness and HC [89], in a recent systematic review and meta-analysis of a pooled sample of patients with MDD and comorbid mental illnesses, Nikolova and colleagues (2021) observed significant differences in alpha diversity relative to HC [92]. Equivocal alpha diversity findings have also been reported in neuropsychiatric diseases with similar pathophysiologies to depression, including Parkinson’s disease [93], autism spectrum disorder [94], and anxiety [95].

We observed a significant difference in beta diversity between patients with MDD and HC across most studies. Beta diversity is a measure of between-samples diversity [95], which in our case, is the similarity of microbial communities in patients with MDD compared to HC. Our observations are consistent with findings in patients with MDD [47], and psychiatric disorders more generally where changes in beta diversity have been noted [92], and aligns with the hypothesis that MDD involves a dysbiotic state [96].

In patients with depression, relative to HC, we also observed several taxa abundance changes. Specifically, there was an increase in the abundance of *Streptococcaceae* and *Bifidobacteriaceae* and families as well as *Eggerthella* and *Streptococcus* genera. Cheung and colleagues (2019) also observed an increase in the genus *Streptococcus* in patients with MDD [47]. *Streptococcus* is a high metabolizer of amino acids and proteins, which may divert essential host amino acids to the microbes in a process called putrefaction and result in toxic products such as ammonia, putrescine, and phenol [97]. Dysbiosis from putrefaction has also been noted in colorectal cancer and autism spectrum disorder [98]. An association of *Eggerthella* with increased gastrointestinal inflammation has also been noted in patients with MDD, and further supports the idea that depression involves inflammatory states [95,99].

*Bifidobacterium* species have been found to have anti-inflammatory effects on stress and depression [100], and are expected to be decreased in MDD. However, our findings of increased abundance are consistent with those noted in a recent systematic review of human studies published from January 2000 to June 2019 on the GMB in depression by Barandouzi and colleagues (2020) [101]. Notably, there is considerable species heterogeneity within taxa, including at the family level, and changes in *Bifidobacteriaceae* levels may not necessarily reflect increases in *Bifidobacterium* species.

We also observed a decrease in the relative abundance of Bacteroidetes phylum, *Sutterellaceae* family, as well as *Coprococcus* and *Faecalibacterium* genera. These results are consistent with the findings from three other reviews [43,47,92]. Nikolova and colleagues (2021), observed a decrease in *Coprococcus* and *Faecalibacterium* across several psychiatric disorders [92]. These genera have anti-inflammatory properties, and are involved in the production of butyrate, a short chain fatty acid involved in the maintenance of the gastrointestinal mucosa and reduction in pro-inflammatory cytokines [102,103]. Mucosal integrity is important for preventing endotoxins from entering the circulation and reducing uncontrolled inflammation [36]. Barandouzi and colleagues also reported similar reductions in the level of *Sutterellaceae* in patients with MDD, but inconsistent findings in the abundance of Bacteroidetes [101]. This observation may be due to the fact that the latter taxon is at the phylum level and therefore encompasses a more heterogeneous set of species. Decreased levels of Bacteroidetes have also been observed in females compared to males, and females were overrepresented among patients with MDD relative to HC in the observational studies [104].

### 3.2. Interventional Trials

When analyzed together, the interventional trials show a modest benefit of probiotic and synbiotic, but not prebiotic treatment in reducing depressive symptoms of patients with MDD over four to nine weeks relative to placebo. Ten probiotic studies with a combined sample size of 543 participants, five of which were double-blind RCTs and four of which were open-label trials, demonstrated a significant decrease in depressive symptoms over time relative to placebo or antidepressant medication. However, seven high quality studies (N = 462), which were either double-blind or triple-blind RCTs, demonstrated no significant changes in depressive symptoms over time compared to placebo. None of the prebiotic studies demonstrated significant changes in depressive symptoms following intervention. The evidence therefore supports some benefit of probiotic, and synbiotic treatment in patients with MDD relative to placebo, but is largely equivocal.

Our observations parallel those of recent reviews on the gut microbiota in MDD. In a recent systematic review and meta-analysis, Hofmeister and colleagues (2021) synthesized evidence from 50 RCTs that evaluated probiotic, prebiotic, synbiotic, paraprobiotic, or fecal microbiota transplant interventions in an adult population [48]. The authors reported statistically significant benefits of probiotic, prebiotic, and synbiotic interventions on depressive symptoms, as measured by the BDI and the depression subscale of the Hospital Anxiety and Depression Scale [48]. These findings are supported by those of an earlier systematic review and meta-analysis of 16 RCTs by El Dib and colleagues (2021), which demonstrated significant improvement in depression and anxiety symptoms in patients treated with probiotics according to the BDI and State-Trait Anxiety Inventory (STAI) [50]. However, in another systematic review of nine randomized double and triple blind placebo-controlled clinical trials, Minayo and colleagues (2021) reported no definitive effect of probiotics on depression and anxiety [49]. Therefore, while there appears to be some benefit of probiotic, prebiotic, and synbiotic treatment in reducing depressive symptoms at 4 to 9 weeks follow up, the evidence to support this observation is mixed.

One possible explanation for these mixed findings, is that the majority of interventional trials published to date include patients who meet strict criteria for MDD. It is possible, however, that patients with mild depression may derive more benefit from probiotic and synbiotic treatment than those with chronic, treatment-resistant depression [53]. Additional studies in subsamples of patients with depression would be helpful in elucidating the benefits of these treatments.

Prebiotic and synbiotic treatments for depression are also largely understudied, and the evidence is less concrete. While some reviews conclude that prebiotic supplementation, either alone or with probiotics, can have beneficial effects on mental health disorders [48,105,106], others report that they do not improve depression or anxiety symptoms [107]. Very few studies have been conducted on these supplements and there is a need for multiple studies on each compound to be performed and analyzed separately.

Regarding potential mechanisms, two studies suggested that changes in depressive symptoms may be due to altered GMB composition as a result of supplementation. There was an increase in the relative abundance of *Bifidobacterium* in the fecal microbiota of patients supplemented with *Bifidobacterium breve CCFM1025* over four weeks [85]. Supplementation with OMNi-BiOTiC^®^ Stress Repair and *Bifidobacterium breve CCFM1025* over four weeks also resulted in increase in *Coprococcus* [80] and *Faecalibacterium* [85] genera, which are normally decreased in MDD. These findings suggest that supplementation may alter GMB pharmacokinetics and pharmacodynamics, and depressive symptoms in patients with depression. 

We did not observe any appreciable changes in the alpha or beta diversity of the GMB, following supplementation, however. Scholars have noted that changes in diversity are sometimes not observed despite the presence of a highly divergent community composition [108], and understanding how alterations in composition affect microbiota functioning may provide more insight into disease [109].

### 3.3. Limitations

We identify important limitations of the studies included in this synthesis. Studies did not consistently control for geographic region or diet, despite these being well-established confounders in the GMB literature [47,110,111]. High fat diets can, for example, increase the concentration of lipopolysaccharides, and stimulate the immune system [112], affecting the gut–brain axis. Moreover, while antidepressant medications, including tricyclic antidepressants (TCAs), monoamine oxidase inhibitors (MAOIs), and selective serotonin reuptake inhibitors (SSRIs), have been found to have antimicrobial effects, contributing to GMB changes, including dysbiosis [113,114], few studies included drug-naïve patients during the time of the intervention. Indeed, one eight-week pilot trial of a probiotic in treatment-naïve patients with depression (which was included in this review) was associated with significant improvements in affective clinical symptoms at four and eight weeks follow-up [53].

An additional limitation of this synthesis is that we analyzed the overall effect of various probiotics, prebiotics, and synbiotics. Ideally, multiple studies for each specific compound should be performed and analyzed separately. These studies would discern whether specific compounds influence depressive symptoms while others do not. It would also be valuable to understand the pharmacokinetics, pharmacodynamics, and mechanism of action of these supplements. For example, do the probiotics engraft in the gut (pharmacokinetics) and lead to a change in microbial composition and metabolites (pharmacodynamics) with a direct effect on mood? Additional studies are needed to determine these effects.

Sex differences in the GMB may also affect the results of the observational studies. Relative to HC, the sample of patients with MDD was predominantly female, who have been shown to have different microbiome profiles than males [104]. Our analysis also focused on gut microbial composition (pharmacokinetics) rather than function (pharmacodynamics). However, functional potentials, including changes in short-chain fatty acid synthesis, tryptophan metabolism [68], and neurotransmitter synthesis and degradation [26,115,116], provide essential insights into the mechanisms of psychiatric illness.

There was considerable heterogeneity in the species and dosage administered in the prebiotic, probiotic, and synbiotic interventions. Additionally, the follow-up time for the majority of studies was short (four to nine weeks). One open-label trial by Bambling and colleagues (2017) included in our analysis, found that there was a significant decrease in BDI score in the probiotic group over eight weeks, but these results did not persist at 16 weeks follow-up [71]. While antidepressants tend to see benefit after six weeks [117], the ideal duration for observing benefits of probiotic treatments is unknown [118]. The benefits of these treatments relative to antidepressants have also not been extensively studied. Therefore, additional studies with continuous monitoring, longer follow-up times, and antidepressant controls are required to elucidate the optimal species, dosage, and treatment length necessary for prebiotic, probiotic and synbiotic interventions to be used in clinical practice.

## 4. Methods

### 4.1. Information Sources and Search Strategy

The systematic review was conducted according to the PRISMA 2020 Guidelines [54]. We conducted two independent searches for observational studies and clinical trials in PubMed, EMBASE, and PsycINFO from 1 January 2016 up to 6 January 2022. 

The search strategy for observational studies was as follows:Depression.mp. or exp Depression/or exp Depressive Disorder, Major/or depressive.mp. or exp Depressive Disorder/or exp Depressive Disorder, Treatment-Resistant/Gastrointestinal microbiome.mp. or exp Gastrointestinal Microbiome/or gut.mp. or fecal.mp. or microbiota.mp. or exp Microbiota/or microbiome.mp.(healthy control or healthy or control).mp.(alpha diversity or beta diversity or abundance* or diversity or rRNA).mp.1 and 2 and 3 and 4Limit 6 to English language, human, 2016-current.

The search strategy for clinical trials was as follows: Depression.mp. or exp Depression/or exp Depressive Disorder, Major/or depressive.mp. or exp Depressive Disorder/or exp Depressive Disorder, Treatment-Resistant/probiotic.mp. or exp Probiotics/or prebiotic.mp. or exp Prebiotics/or synbiotic.mp. or exp Synbiotics/or exp Lactobacillus/or Lactobacillus.mp. or exp Bifidobacterium/exp Clinical Trial/or trial.mp.1 and 2 and 3Limit 5 to English language, human, 2016–current.

We also identified additional articles for inclusion by searching the full-texts of reviews conducted on the GMB and depression.

### 4.2. Eligibility

Observational studies that met the following criteria were included: (1) they were conducted in patients with a diagnosis of MDD according to a validated scale, (2) they involved healthy controls, (3) they conducted GMB analysis (taxa abundance differences, alpha diversity or beta diversity), and (4) they were peer-reviewed and published as full-texts in English. 

Clinical trials that met the following criteria were included: (1) they were conducted in patients with a diagnosis of MDD, depressive episode or depressive symptoms according to a validated scale, (2) they involved a prebiotic, probiotic, or symbiotic intervention, and (3) they were peer-reviewed and published as full-texts in English.

For both observational studies and clinical trials, we excluded (1) studies published before 2016, (2) reviews, case reports, conference abstracts, dissertations, or letters, (3) protocol descriptions of studies not yet conducted, and (4) studies reporting only pooled results in patients with comorbid psychiatric and medical conditions (e.g., bipolar disorder (BD), schizophrenia (SZ), anxiety, and irritable bowel syndrome (IBS)). Studies involving patients with comorbidities were included as long as they conducted subgroup analyses in patients with MDD alone.

### 4.3. Selection Process

The first author (S.R.A.) screened all records (titles and abstracts) and reports (full-texts). In cases of uncertainty, articles were independently screened by D.J.M. and I.G. until consensus was reached. Original study investigators were not contacted. No automation tools were used.

### 4.4. Outcome Measures and Data Items

For the observational studies, the primary outcomes of interest were differences in GMB composition between patients with depression and healthy controls, as measured by alpha diversity, beta diversity, and taxa abundance changes. 

Alpha diversity is a measure of the richness (number of species) and evenness (distribution) of the microbial community within one sample [88]. It is assessed by several measures, including the Shannon index, Simpson index, phylogenetic diversity, total observed species or operational taxonomic units (OTUs), Chao 1, Inverse Simpson index, Sobe index and Abundance-based Coverage Estimator (ACE) [119]. 

Beta diversity, on the other hand, is a measure of the similarity or dissimilarity between the microbiota communities of two samples [120]. It can be measured by simple taxa overlap, the Bray–Curtis Dissimilarity index, and UniFrac distance, among other indices [88]. A significant difference in beta diversity between two groups indicates that the communities have significantly different species composition. Therefore, alpha diversity measure represents a summary statistic of a single population, while beta diversity measure represents a similarity score between populations.

Taxa abundance changes reflect the proportion of bacteria at domain, kingdom, phylum, class, order, family, and genus levels in one group relative to another (for our purposes, patients with MDD compared to HC). Given inconsistent reporting about species, we did not assess changes at the species level.

For clinical trials, the primary outcome measures were rating scale scores for depressive symptoms at four to nine weeks follow-up. Where multiple measures of depressive symptoms were available, we reported the results of validated, commonly used rating scales. We also collected information on microbiota changes (alpha diversity, beta diversity, and taxa abundance) as a secondary outcome.

Data items that were collected for all studies include study design, mean age and standard deviation, sex, country, population, sample size, diagnostic definition of depression, depression rating scale, and genetic analysis techniques. Specifically for the clinical trials, we also collected data on the intervention type, control groups, trial length, probiotic bacterial strains, and depressive symptom score changes (see Table 1 and Table 2).

## 5. Conclusions

Taken together, the findings from observational studies conducted between January 2016–January 2022 provide evidence for a specific gut microbial profile of patients with MDD compared to HC. The interventional trials suggest that there is modest benefit of probiotic and synbiotic, but not prebiotic, supplementation in reducing the symptoms of depression relative to placebo, and that probiotic treatment may influence GMB composition. However, additional and more rigorous double-blind randomized-controlled trials, which consider confounding factors such as symptom severity, age, diet, and medication use, are needed. Critical questions about species administered, dosage, and length of treatment remain to be addressed before these therapies reach the implementation stage as treatments for depression.

## Figures and Tables

**Figure 1 ijms-23-04494-f001:**
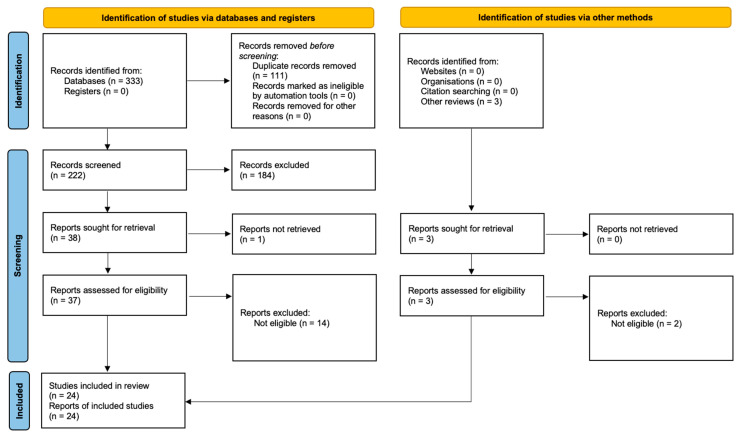
Modified Prisma Flow Diagram [54] for the observational studies on gut microbiome changes in patients with major depressive disorder included in the systematic review.

**Figure 2 ijms-23-04494-f002:**
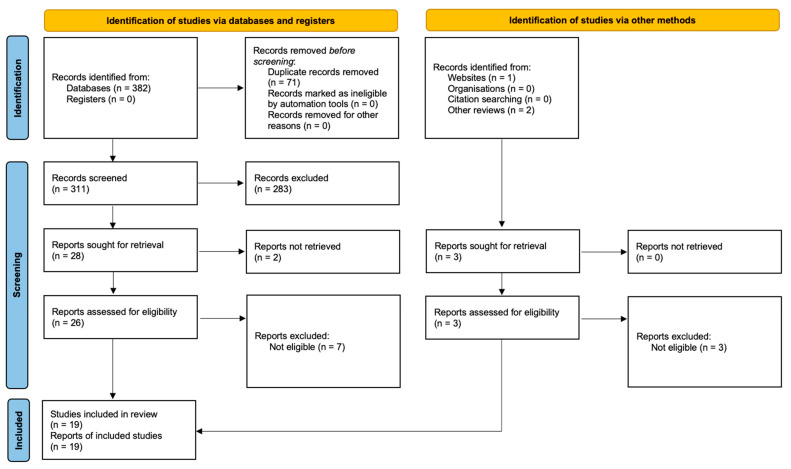
Modified Prisma Flow Diagram [54] for the clinical trials on prebiotics, probiotics and synbiotics in patients with major depressive disorder included in the systematic review.

**Table 1 ijms-23-04494-t001:** Characteristics of observational studies from the past five years included in the systematic review.

Study	Study Design	Country	Population	Definition of Depression	Mean Age (SD)	Sex (%F)	Outcomes
Aizawa 2016 [55]	Cross-Sectional	Japan	MDD (N = 43) HC (N = 57)	DSM-IV	MDD: 41.9 HC: 61.4	MDD: 41.9 HC: 61.4	Fecal microbiota One time
Kelly 2016 [22]	Cross-Sectional	Ireland	MDD (N = 34) HC (N = 33)	DSM-IV HAM-D ≥ 17	MDD: 45.8 (11.5) HC: 45.8 (11.9)	MDD: 32.8 HC: 42.4	Fecal microbiota One time
Liu 2016 [56]	Cross-Sectional	China	MDD (N = 15) IBS-D (N = 40) COMO (N = 25) HC (N = 33)	MINI DSM-IV	MDD: 73.3 IBS-D: 30.0 COMO: 44.0 HC: 65.0	MDD: 73.3 IBS-D: 30.0 COMO: 44.0 HC: 65.0	Fecal microbiota One time Sigmoid mucosa
Zheng 2016 [57]	Cross-Sectional	China	MDD (N = 58) HC (N = 63)	DSM-IV HAM-D	MDD: 62.1 HC: 63.5	MDD: 62.1 HC: 63.5	Fecal microbiota One time
Lin 2017 [58]	Prospective	China	MDD (N = 10) HC (N = 10)	DSM-IV HAM-D ≥ 23	MDD: 36.2 (10.1) HC: 38.1 (2.9)	MDD: 60.0 HC: 60.0	Fecal microbiota Three times over one month
Chen 2018a [62]	Cross-Sectional	China	MDD (N = 44) HC (N = 44)	HAM-D	MDD: 40.9 (11.2) HC: 43.4 (13.4)	MDD: 54.5 HC: 54.5	Fecal microbiota One time
Chen 2018b [61]	Cross-Sectional	China	MDD (N = 10) HC (N = 10)	DSM-IV HAM-D ≥ 20	MDD: 43.9 (13.8) HC: 39.6 (9.0)	MDD: 50.0 HC: 50.0	Fecal microbiota One time
Huang 2018 [59]	Cross-Sectional	China	MDD (N = 27) HC (N = 27)	ICD-10	MDD: 48.7 (12.8) HC: 42.3 (14.1)	MDD: 74.0 HC: 74.0	Fecal microbiota One time
Chung 2019 [23]	Cross-Sectional	Taiwan	MDD (N = 36) HC (N = 37)	DSM-IV	MDD: 45.83 (14.08) HC: 41.19 (12.73)	MDD: 82.35 HC: 62.16	Fecal microbiota One time
Rong 2019 [28]	Cross-Sectional	China	MDD (N = 31) BD-D (N = 30) HC (N = 30)	DSM-V	MDD: 41.58 (10.40) BD-D: 38.40 (8.33) HC: 39.47 (10.22)	MDD: 70.97 BD-D: 50.00 HC: 53.33	Fecal microbiota One time
Chen 2020 [25]	Cross-Sectional	China	Y-MDD (N = 25) Y-HC (N = 27) M-MDD (N = 45) M-HC (N = 44)	DSM-IV	Y-MDD: 24.0 (3.74) Y-HC: 24.96 (2.31) M-HC: 47.16 (8.07) M-MDD: 44.96 (7.76)	Y-HC: 70.37 Y-MDD: 72.0 M-HC: 77.2 M-MDD: 68.89	Fecal microbiota One time
Liu 2020 [24]	Cross-Sectional	USA	MDD (N = 43) HC (N = 47)	SCID-4	MDD: 21.9 (2.1) HC: 22.1 (1.8)	MDD: 88.4 HC: 72.3	Fecal microbiota One time
Mason 2020 [30]	Cross-Sectional	USA	MDD (N = 38) -Anxiety only (N = 8) -Depression only (N = 14) HC (N = 10)	SCID-5	MDD: 39.2 -Anxiety only (40.0) -Depression only (41.9) HC: 33	MDD: 82 -Anxiety only: 100 -Depression only: 79 HC: 60	Fecal microbiota One time
Rhee 2020 [29]	Cross-Sectional	South Korea	MDD (N = 30) BD (N = 42) HC (N = 36)	DSM-V MINI	MDD: 46.2 (9.7) BD: 34.2 (10.8) HC: 43.0 (5.6)	MDD: 83.3 BD: 64.3 HC: 75.0	Fecal microbiota One time
Yang 2020 [26]	Cross-Sectional	China	MDD (N = 156) HC (N = 155)	DSM-IV MINI	D-HC: 26.86 (5.24) D-MDD: 27.19 (4.71) V-HC: 36.39 (10.75) V-MDD: 37.07 (9.45)	D-HC: 56.78 D-MDD: 56.78 V-HC: 64.86 V-MDD: 86.84	Fecal microbiota
Zheng 2020 [69]	Case-Control	China	MDD (N = 165) BD (N = 217) HC (N = 217)	DSM-IV	MDD: 26.54 (4.07) BD: 25.59 (8.41) HC: 26.85 (5.48)	MDD: 63.11 BD: 49.70 HC: 58.48	Fecal microbiota One time
Bai 2021 [66]	Cross-Sectional	China	MDD (N = 60) HC (N = 60)	DSM-IV HAM-D > 17	MDD: 35.62 (17.10) HC: 35.13 (15.79)	HC: 60.0 MDD: 65.0	Fecal microbiota One time
Caso 2021 [67]	Cross-Sectional	Spain	a-MDD (N = 46) r-MDD (N = 22) HC (N = 45)	DSM-IV HAM-D > 14	a-MDD: 42.10 r-MDD: 45.85 HC: 44.72	a-MDD: 78.26 r-MDD: 77.27 HC: 75.5	Fecal microbiota One time
Chen 2021 [27]	Cross-Sectional	China	MDD (N = 62) HC (N = 46)	DSM-V MINI HAMD-17 ≥ 18	HC: 36.93 (8.58) MDD: 39.58 (12.66)	ND	Fecal microbiota One time
Dong 2021 [65]	Cross-Sectional	China	MDD (N = 23) GAD (N = 21) HC (N = 10)	DSM-V	MDD: 30.04 (5.90) GAD: 30.43 (7.95) HC: 30.22 (6.50)	MDD: 69.57 GAD: 66.67 HC: 60.00	Fecal microbiota One time
Lai 2021 [68]	Cross-Sectional	China	MDD (N = 26) HC (N = 29)	SCID-V HAM-D > 17	MDD: 43.73 (11.46) HC: 39.41 (10.96)	MDD: 69.2 HC: 55.2	Fecal microbiota One time
Thapa 2021 [60]	Longitudinal	USA	MDD (N = 110) HC (N = 27) Psy ctr (N = 23)	DSM-IV-TR	MDD: 19.5 (0.4) HC: 20.3 (0.2) Psy ctr: 19.1 (0.4)	MDD: 65 HC: 37 Psy ctr: 43	Fecal microbiota One time
Zhang 2021 [64]	Case-Control	China	MDD (N = 36) HC (N = 45)	ICD-10	MDD: 36.81 (13.52) HC: 39.29 (11.44)	MDD: 41.7 HC: 57.8	Fecal microbiota One time
Zheng 2021 [63]	Case-Control	China	MDD (N = 30) HC (N = 30)	ICD-10	MDD: 30.80 (10.85) HC: 33.37 (7.02)	MDD: 60.0 HC: 56.7	Fecal microbiota One time

*Note.* Abbreviations: a-MDD, active-major depressive disorder; r-MDD, Major depressive disorder in remission or with only mild symptoms; BD-D, Bipolar disorder in current depressive episode; BD, Bipolar disorder; COMO, comorbid irritable bowel syndrome and depression; D-HC; Discovery set of healthy controls; D-MDD, Discovery set of patients with major depressive disorder; DSM-IV-TR, Diagnostic and Statistical Manual of Mental Disorders-Fourth Edition-Text Revision; DSM-IV, Diagnostic and Statistical Manual of Mental Disorders-Fourth Edition; DSM-V, Diagnostic and Statistical Manual of Mental Disorders-Fifth Edition; HAM-D, Hamilton Depression Rating Scale; IBS-D, Irritable bowel syndrome–Diarrhea predominant; ICD-10, International Statistical Classification of Diseases and Related Health Problems-10; MDD, Major depressive disorder; M-HC, Middle-aged healthy controls; M-MDD, Middle-aged major depressive disorder; MDD, major depressive disorder; Psy ctr, Psychiatric controls; SCID-4, Structured Clinical Interview for DSM-4; SCID-5, Structured Clinical Interview for DSM-5; SD, standard deviation; V-HC, Validation set of healthy controls; V-MDD, Validation set of patients with major depressive disorder; Y-HC, Young healthy controls; Y-MDD, Young major depressive disorder.

**Table 2 ijms-23-04494-t002:** Alpha diversity and beta diversity changes observed in patients with major depressive disorder relative to healthy controls in the observational studies.

Study	Genetic Analysis	Alpha Diversity Findings ^a^	Beta Diversity Findings ^b^
Aizawa 2016 [55]	Analysis: 16S rRNA sequencing Platform: Yakult Intestinal Flora-SCAN^®^	ND	ND
Kelly 2016 [22]	Analysis: 16S rRNA sequencing Platform: Illumina MiSeq Region: ND Pipeline: QIIME Database: SILVA	Significant decrease in MDD compared to HC (Chao, observed species, phylogenetic diversity). No significant difference between MDD and HC (Shannon).	No significant difference between MDD and HC (Weighted Bray–Curtis similarity, Unweighted UniFrac distances, Weighted UniFrac distances).
Liu 2016 [56]	Analysis: 16S rRNA sequencing Platform: Roche 454 sequencing Region: V1-V3 Pipeline: Mothur Database: RDP	No significant difference between MDD and HC (Shannon).	ND
Zheng 2016 [57]	Analysis: 16S rRNA sequencing Platform: Roche 454 sequencing Region: V3-V5 Pipeline: Mothur Database: RDP	No significant difference between MDD and HC (observed species, phylogenetic diversity, Shannon, Simpson).	Significant difference between MDD and HC (Weighted Bray–Curtis similarity, Unweighted UniFrac distances).
Lin 2017 [58]	Analysis: 16S rRNA sequencing Platform: Illumina MiSeq Region: V3-V4 Pipeline: Mothur Database: SILVA	ND	No significant difference between MDD and HC (Weighted UniFrac distances).
Chen 2018a [62]	Analysis: 16S rRNA sequencing Platform: Roche 454 sequencing Region: V3-V5 Pipeline: Mothur Database: RDP	No significant difference between MDD and HC (phylogenetic diversity).	Significant difference between MDD and HC (UniFrac distances, PLS-DA).
Chen 2018b [61]	Analysis: Metaproteomics	ND	ND
Huang 2018 [59]	Analysis: 16S rRNA sequencing Platform: Illumina HiSeq Region: V3-V4 Pipeline: QIIME Database: GreenGenes	Significant decrease in MDD compared to HC (ACE, Chao, phylogenetic diversity, Shannon).	No significant difference between MDD and HC (Unweighted UniFrac distances, Weighted UniFrac distances).
Chung 2019 [23]	Analysis: 16S rRNA sequencing Platform: Illumina MiSeq Region: V3-V4 Pipeline: QIIME Database: GreenGenes	No significant difference between MDD and HC (Chao, observed OTUs, phylogenetic diversity, Shannon).	Significant difference between MDD and HC (Unweighted UniFrac distances, Weighted UniFrac distances).
Rong 2019 [28]	Analysis: SMS Platform: Illumina HiSeq Database: KEGG	Significant decrease in MDD compared to HC (Chao). No significant difference between MDD and HC (Inverse Simpson, Shannon).	ND (MDD vs. HC)
Chen 2020 [25]	Analysis: 16S rRNA sequencing Platform: Roche 454 sequencing Region: V3-V5 Pipeline: Mothur Database: RDP	No significant difference between MDD and HC (ACE, Chao).	ND
Liu 2020 [24]	Analysis: 16S rRNA sequencing Platform: Illumina MiSeq Region: V4 Pipeline: QIIME2 Database: SILVA	Significant decrease in MDD compared to HC (phylogenetic diversity). No significant difference between MDD and HC (Shannon, Simpson, observed ASVs).	Significant difference between MDD and HC (Unweighted UniFrac distances, Bray–Curtis). No significant difference between MDD and HC (Weighted UniFrac distances).
Mason 2020 [30]	Analysis: 16S rRNA sequencing Platform: Roche 454 sequencing Region: V4 Pipeline: QIIME Database: SILVA	No significant difference between MDD and HC (Shannon).	No significant difference between MDD and HC (Weighted UniFrac distance).
Rhee 2020 [29]	Analysis: 16S rDNA sequencing Platform: Illumina MiSeq Region: V3-V4 Pipeline: QIIME Database: SILVA	Significant increase in MDD compared to HC (Inverse Simpson, Shannon). No significant difference between MDD and HC (Chao, observed OTUs).	Significant difference between MDD and HC (Unweighted UniFrac distances, Bray–Curtis). No significant difference between MDD and HC (Weighted UniFrac distances).
Yang 2020 [26]	Sequencing: SMS Platform: Illumina NovaSeq Database: KEGG, NCBI NR	No significant difference between MDD and HC (Chao, Shannon, Simpson, Inverse Simpson).	Significant difference between MDD and HC (Bray–Curtis Distance).
Zheng 2020 [69]	Analysis: 16S rRNA sequencing Platform: Illumina MiSeq Region: V3-V4 Pipeline: UPARSE Database: RDP	No significant difference between MDD and HC (Ace, Chao, Shannon, Inverse Simpson).	Significant difference between MDD and HC (PLS-DA).
Bai 2021 [66]	Analysis: 16S rRNA sequencing Platform: ND Region: ND Pipeline: ND Database: RDP	No significant difference between MDD and HC (Chao, Shannon, Simpson, phylogenetic diversity).	Significant difference between MDD and HC (PCoA).
Caso 2021 [67]	Analysis: 16S rDNA sequencing Platform: Illumina MiSeq Region: V3-V4 Pipeline: QIIME, Calypso Database: RDP	No significant difference between MDD and HC (Shannon).	No significant differences between MDD and HC (Bray–Curtis, Binary Jaccard).
Chen 2021 [27]	Analysis: 16S rRNA sequencing Platform: Illumina MiSeq Region: V3-V4 Pipeline: Mothur, QIIME Database: RDP	No significant difference between MDD and HC (ACE, Chao, Shannon, Simpson,).	Significant difference between MDD and HC (Weighted UniFrac, Unweighted UniFrac).
Dong 2021 [65]	Analysis: 16S rRNA sequencing Platform: Illumina MiSeq Region: V3-V4 Pipeline: QIIME2 Database: SILVA	No significant difference between MDD and HC (ACE, Chao, Shannon, Simpson).	No significant difference between MDD and HC (Bray–Curtis).
Lai 2021 [68]	Analysis: SMS Platform: Illumina HiSeq Database: KEGG	Significant difference between MDD and HC (Fisher). No significant difference between MDD and HC (Shannon).	Significant difference between MDD and HC (PCoA).
Thapa 2021 [60]	Analysis: 16S rRNA sequencing Platform: Illumina MiSeq Region: V4 Pipeline: QIIME Database: SILVA	No significant difference between MDD and HC (ACE, Chao, Observed OTUs, phylogenetic diversity, Shannon).	No significant difference between MDD and HC (Bray–Curtis, Unweighted UniFrac distances, Weighted UniFrac distances, Aitchison distance).
Zhang 2021 [64]	Analysis: 16S rRNA sequencing Platform: Illumina Region: V4-V5 Pipeline: Mothur, UPARSE, R Database: ND	No significant difference between MDD and HC (ACE, Chao, Shannon, Simpson).	No significant difference between MDD and HC (Unweighted UniFrac distances, Weighted UniFrac distances).
Zheng 2021 [63]	Analysis: 16S rRNA sequencing Platform: Illumina MiSeq Region: ND Pipeline: Mothur Database: ND	No significant difference between MDD and HC (ACE, Chao, Shannon, Simpson).	ND

*Note.* Abbreviations: ACE, Abundance-based Coverage Estimator; ASV, Amplicon sequence variant; KEGG, Kyoto Encyclopedia of Genes and Genomes; NCBI NR, National Center for Biotechnology Information Non-Redundant Database; ND, Not declared; OTU, Operational taxonomic unit; PCoA, Principal coordinate analysis; PLS-DA, Partial least squares discriminant analysis; QIIME, Quantitative Insights into Microbial Ecology; RDP, Ribosomal Database Project; SMS, Shotgun metagenomics sequencing. ^a^ Microbial community composition differences within groups. ^b^ Microbial community composition differences between groups.

**Table 3 ijms-23-04494-t003:** Summary of taxa abundance changes in patients with major depressive disorder relative to healthy controls ^a^.

Taxon	Increased in MDD	Decreased in MDD
Phylum		Bacteroides
Family	*Bifidobacteriaceae*	*Sutterellaceae*
	*Streptococcaceae*	
Genus	*Eggerthella*	*Coprococcus*
	*Streptococcus*	*Faecalibacterium*

^a^ Taxa abundance changes observed in four or more studies are presented here.

**Table 4 ijms-23-04494-t004:** Characteristics of the clinical trials on prebiotics, probiotics, and synbiotics in major depressive disorder included in the systematic review.

Authors	Study Design	Country	Population	Depression Definition	Mean Age (SD)	Sex (% F)
Akkasheh 2016 [70]	DB RCT	Iran	MDD (N = 40)	DSM-IV; HAMD-17 ≥ 15	Pro: 38.3 (12.1) Plb: 36.2 (8.2)	ND
Bambling 2017 [71]	Open-label trial	Australia	Resistant MDD (N = 12)	MINI-V	Pro: 49.3 (10.9)	Pro: 66.7
Romijn 2017 [72]	DB RCT	New Zealand	Low mood (N = 79)	QIDS-SR16 ≥ 11; DASS-42-D ≥ 14	Pro: 35.8 (14) Plb: 35.1 (14.5)	Pro: 20 Plb: 23
Ghorbani 2018 [73]	DB RCT	Iran	MDD (N = 40)	DSM-V	Syn: 34.45 Plb: 35.50	Syn: 70 Plb: 70
Miyaoka 2018 [74]	Prospective open-label trial	Japan	TRD (N = 40)	DSM-IV-TR	Pro: 44.2 (15.6) Ctr: 41.9 (14.2)	Pro: 52.0 Ctr: 52.0
Chahwan 2019 [77]	TB RCT	Australia	Clinical and sub-clinical depression (N = 71)	MINI-IV	Pro: 36.65 (11.75) Plb: 35.49 (12.34) Ctr: 35.95 (11.74)	Pro: 21 Plb: 28 Ctr: 15
Kazemi 2019 [75]	DB RCT	Iran	MDD (N = 110)	ICD-10	Pro: 36.2 Pre: 75.0 Plb: 66.7	Pro: 71.1 Pre: 75.0 Ctr: 66.7
Rudzki 2019 [76]	DB RCT	Poland	MDD (N = 60)	DSM-IV-TR	Pro: 39.13 (9.96) Plb: 38.90 (12)	Pro: 76.7 Plb: 66.7
Heidarzadeh 2020 [79]	DB RCT	Iran	MDD (N = 78)	ICD-10	Pro: 36.2 Pre: 75.0 Plb: 66.7	Pro: 71.1 Pre: 75.0 Plb: 66.7
Reininghaus 2020 [80]	DB RCT	Austria	MDD (N = 82)	MINI-IV	Pro: 43.00 (14.31) Plb: 40.11 (11.45)	Pro: 71.4 Plb: 81.8
Reiter 2020 [81]	DB RCT	Austria	MDD (N = 61)	MINI-IV	Pro: 43.00 (14.31) Plb: 40.11 (11.45)	Pro: 71.4 Plb: 81.8
Saccarello 2020 [82]	DB RCT	Italy	MDD (N = 90)	ICD-10	Pro: 48.6 (10.67) Plb: 47.5 (11.9)	Pro: 84.4 Plb: 75
Arifdjanova 2021 [87]	Open RCT	Russia	Mild-moderate depressive episode	ICD-10	Pro: 32.9 (6.1) Plb: 33.1 (5.7)	Pro and Plb: 62.2
Browne 2021 [83]	DB pilot trial	Netherlands	Depressive sxs (N = 40)	EPDS ≥ 10	Pro: 29.65 (3.9) Plb: 31.7 (4)	ND
Chen 2021 [84]	Open trial	Taiwan	MDD (N = 11)	DSM-V	Pro: 39.4 (12.0)	Pro: 72.7
Vaghef-Mehrabany 2021 [78]	DB RCT	Iran	MDD (N = 62)	DSM-V	Pre: 37.45 Plb: 40.00	ND
Wallace 2021 [53]	Open pilot study	Canada	MDD (N = 10)	MINI-IV; MADRS ≥ 20	Pro: 25.2 (7.0) Plb: 40.00	Pro: 70
Zhang 2021 [86]	DB RCT	China	MDD (N = 69)	DSM-V	Pro: 45.8 (12.3) Plb: 49.7 (9.6)	Pro: 63.2 Plb: 84.5
Tian 2022 [85]	DB RCT	China	MDD (N = 45)	HAMD-24 > 14	Pro: 51.32 (16.11) Plb: 48.15 (13.96)	Pro: 70.0 Plb: 64.0

*Note.* Abbreviations: CES-D, Center for Epidemiological Studies Depression Scale; Ctr, control; DASS-42-D, depression subscale of the Depression, Anxiety and Stress Scale; DB, double-blind; DSM-IV-TR, Diagnostic and Statistical Manual of Mental Disorders-Fourth Edition-Text Revision; DSM-IV, Diagnostic and Statistical Manual of Mental Disorders-Fourth Edition; DSM-V, Diagnostic and Statistical Manual of Mental Disorders-Fifth Edition; EPDS, Edinburgh Postnatal Depression Scale; HAM-D, Hamilton Depression Rating Scale; HAMD-17, Hamilton Depression Rating Scale-17; HAMD-24, Hamilton Depression Rating Scale-24; ICD-10, International Statistical Classification of Diseases and Related Health Problems-10; MADRS, Montgomery-Asberg Depression Rating Scale; MDD, major depressive disorder; MINI-IV, Mini-International Neuropsychiatric Interview for DSM-IV; MINI-V, Mini-International Neuropsychiatric Interview for DSM-V; ND, Not declared; Plb, placebo; Pre, prebiotic; Pro, probiotic; QIDS-SR16, Quick Inventory of Depressive Symptomatology; RCT, Randomized controlled trial; SB, single-blind; SSRI, Selective serotonin reuptake inhibitor; Sxs, symptoms; Syn, synbiotic; TB, triple-blind; TRD, treatment-resistant major depressive disorder.

**Table 5 ijms-23-04494-t005:** Major findings in the clinical trials on prebiotics, probiotics and synbiotics in major depressive disorder included in the systematic review.

Authors	Population	Intervention	Control	Trial Length	Outcome Measure	Depressive Symptom Score Changes	Microbiome Changes
Akkasheh 2016 [70]	MDD (N = 40)	*L. acidophilus, L. casei, B. bifidum*	Plb	8 weeks	BDI	Significant decrease in BDI score in probiotic group compared to placebo over 8 weeks.	ND
Bambling 2017 [71]	Resistant MDD (N = 12)	Mg2+, *L. acidophilus, B. bifidum, S. thermophiles*	No ctr	8 weeks 8, 16-wk f/u	BDI	Significant decrease in BDI score in probiotic group over 8 weeks, but not at 16 weeks.	ND
Romijn 2017 [72]	Low mood (N = 79)	*L. helveticus R0052, B. longum R0174*	Matched Plb	8 weeks	MADRS	No significant difference in MADRS score in probiotic group compared to placebo over 8 weeks.	ND
Ghorbani 2018 [73]	MDD (N = 40)	Familact H^® a^ Syn, Fluoxetine	Plb, Fluoxetine	8 weeks	HAMD-17	Significant decrease in HAMD-17 score in synbiotic group compared to placebo over 8 weeks.	ND
Miyaoka 2018 [74]	TRD (N = 40)	*C. butyricum miyairi 588*, Antidepressants	Anti-depressants	6 weeks	HAMD-17, BDI	Significant decrease in HAMD-17 and BDI score in probiotic group compared to control over 6 weeks.	ND
Chahwan 2019 [77]	Clinical and sub-clinical depression (N = 71)	Ecologic^®^Barrier ^c^	Plb Ctr	8 weeks	BDI	No significant difference in BDI score between probiotic group and placebo over 8 weeks.	No significant differences in α-diversity or β-diversity between probiotic and placebo groups over time.
Kazemi 2019 [75]	MDD (N = 110)	CEREBIOME^® b^	Plb	8 weeks	BDI	Significant decrease in BDI score in probiotic group compared to placebo or prebiotic over 8 weeks. No significant decrease in prebiotic group BDI score compared to placebo over 8 weeks.	ND
Rudzki 2019 [76]	MDD (N = 60)	*L. Plantarum 299v*, SSRI	Plb + SSRI	8 weeks	HAMD-17	No significant difference in HAMD-17 score between probiotic group and placebo over 8 weeks.	ND
Heidarzadeh 2020 [79]	MDD (N = 78)	CEREBIOME^® b^	Plb	8 weeks	BDI-II	Significant decrease in BDI-II score in probiotic compared to placebo over 8 weeks. No significant difference in BDI-11 score between probiotic and prebiotic, or prebiotic and placebo groups over 8 weeks.	ND
Reininghaus 2020 [80]	MDD (N = 82)	OMNi-BiOTiC^®^ Stress Repair ^d^	Plb (Biotin, B7)	4 weeks	HAM-D, BDI-II	No significant decrease in HAM-D and BDI-II score in probiotic group compared to placebo over 4 weeks.	No significant differences in α-diversity between probiotic and placebo groups over time. β-diversity was significantly different in the probiotics group after 28 days. Increased *Ruminococcus gauvreauii* and *Coprococcus 3* abundance in Pro after 28 days.
Reiter 2020 [81]	MDD (N = 61)	OMNi-BiOTiC^®^ Stress Repair ^d^, Biotin	Plb (Biotin, B7)	4 weeks	HAM-D, BDI-II	No significant decrease in HAM-D and BDI-II score in probiotic group compared to placebo over 4 weeks.	ND
Saccarello 2020 [82]	MDD (N = 90)	SAMe, *L. plantarum HEAL9*	Plb	6 weeks 2,6-wk f/u	Z-SDS	Significant decrease in Z-SDS score in probiotic group compared to placebo at week 2 and 6.	ND
Arifdjanova 2021 [87]	Mild-moderate depressive episode	Bac-Set-Forte ^e^	Plb	6 weeks	HAMD-17	Significant decrease in HAMD-17 score in probiotic group compared to placebo over 6 weeks.	ND
Browne 2021 [83]	Depressive sxs (N = 40)	Ecologic^®^Barrier ^c^	Plb	8 weeks	EPDS	No significant decrease in EDPS score in probiotic group compared to placebo over 8 weeks.	ND
Chen 2021 [84]	MDD (N = 11)	*L. plantarum PS128*	None	8 weeks	HAMD-17	Significant decrease in HAMD-17 score in probiotic group over 8 weeks.	No significant differences in α-diversity and β-diversity in probiotic group over 8 weeks.
Vaghef-Mehrabany 2021 [78]	MDD (N = 62)	Inulin 10 g/day	Plb (Maltodextrin 10 g/day)	8 weeks	HAM-D BDI-II	No significant difference in HAM-D and BDI-II score in prebiotic group compared to placebo over 8 weeks.	
Wallace 2021 [53]	MDD (N = 10)	CEREBIOME^® b^	No ctr	8 weeks	MADRS	Significant decrease in MADRS score in probiotic group between baseline and 4 weeks. No significant decrease between 4 weeks and 8 weeks.	ND
Zhang 2021 [86]	MDD (N = 69)	*L. casei Shirota*	Plb	9 weeks	BDI, HAM-D	No significant decrease in HAM-D score in probiotic group compared to placebo over 9 weeks.	No significant differences in α-diversity and β-diversity between probiotic and placebo groups over 9 weeks. Increased *Adlercreutzia*, *Megasphaera* and *Veillonella* and decreased *Rikenellaceae_RC9_gut_group*, *Sutterella* and *Oscillibacter* in probiotic compared to placebo over 9 weeks.
Tian 2022 [85]	MDD (N = 45)	*B. breve CCFM1025*	Plb (Maltodextrin)	4 weeks	HAMD-17	Significant decrease in HAMD-17 score in probiotic group compared to placebo over 4 weeks.	Significant difference in α-diversity between probiotic and placebo according to Chao 1 index and observed operational taxonomic units (OTUs) but not the Shannon index. No significant difference in β-diversity. Increased *Desulfovibrio, Faecalibacterium* and *Bifidobacterium* in probiotic compared to placebo over 4 weeks.

*Note.* Abbreviations: BDI, Beck Depression Inventory; BDI-II, Beck Depression Inventory-II; CES-D, Center for Epidemiological Studies Depression Scale; Ctr, control; DASS-42-D, depression subscale of the Depression, Anxiety and Stress Scale; F/u, follow up; HAM-D, Hamilton Depression Rating Scale; HAMD-17, Hamilton Depression Rating Scale-17; MADRS, Montgomery-Asberg Depression Rating Scale; MDD, major depressive disorder; ND, Not declared; Plb, placebo; Pre, prebiotic; Pro, probiotic; RCT, Randomized controlled trial; Ref, Reference; SAMe, S-adenosylmethionine; SSRI, Selective Serotonin Reuptake Inhibitor; Syn, synbiotic; Z-SDS, Zhung Self-rating Depression Scale. ^a^ Familact H^®^ synbiotic consists of *L. casaei*, *L. acidofilus*, *L. bulgarigus*, *L. rhamnosus, B. breve*, *B. longum*, and *S. thermophilus*. ^b^ CEREBIOME^®^ consists of *L. helveticus R0052* and *B. longum R0175*. ^c^ Ecologic^®^Barrier consists of *B. bifidum W23*, *B. lactis W51*, *B. lactis W52*, *L. acidophilus W37*, *L. brevis W63*, *L. casei W56*, *L. salivarius W24*, *L. lactis W19* and *L. lactis W58*. ^d^ OMNi-BiOTiC^®^ Stress Repair consists of *B. bifidum W23*, *B. lactis W51*, *B. lactis W52*, *L. acidophilus W22*, *L. casei W56*, *L. paracasei W20*, *L. plantarum W62*, *L. salivarius W24* and *L. lactis W19*. ^e^ Bac-Set-Forte consists of *S. thermophilus*, *B. infantis*, *B. bifidum*, *B. breve*; *B. longum*, *L. delbrueckii*, *L. bulgaricus*, *L. helveticus*, *L. salivarius, L. fermentum*, *L. casei*, *L. plantarum*, *L. rhamnosus*, *L. acidophilus*, and *L. lactis*.

## Data Availability

No new data were created or analyzed in this study. Data sharing is not applicable to this article.

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
