# Peer review of "The Gut Microbiome in Depression and Potential Benefit of Prebiotics, Probiotics and Synbiotics: A Systematic Review of Clinical Trials and Observational Studies"

_ijms, 2022, doi:10.3390/ijms23094494_

Round 1
Reviewer 1 Report
In this review, authors have compiled observational and clinical studies on the effects of different compounds that modify gut microbiome on depression. After following different criteria of eligibility, only 24 observational studies and 19 clinical studies were included in the present review. The type of genetic analyses and the main conclusions from each study have been meticulously included in different tables. However, these tables are very large and the conclusions in the present review do not differ from previous reviews in the field. Authors should emphasize the novelty of their study. In addition, authors state broad conclusions from very detailed data. It seems that authors did not take advantage from all the data that they were able to compile and organize.
Line 304: “The majority of these studies reported no significant differences in alpha and beta diversity following prebiotic, probiotic, and synbiotic administration.” Please, state how many.
According to this statement “(line 340) the observational studies demonstrated no significant differences in alpha diversity in patients with MDD compared to HC.”; Why authors consider that “(line 347) Our findings are consistent with those observed in a systematic review and meta-analysis by Nikolova and colleagues (2021), who observed significant differences in alpha diversity in a pooled sample of patients with various psychiatric disorders.”? Authors did not analyze a pooled sample with various disorders. The statement above (line 340) is confusing.
Line 417: “Therefore, while there appears to be some benefit of probiotic, prebiotic, and synbiotic treatment in reducing depressive symptoms at 4 to 9 weeks follow up, the evidence to support this observation is mixed.” Authors should investigate further these studies and find the differences between them that lead to opposite conclusions.
According to the limitations of most the compiled studies, have authors identified what are the better designed and performed studies? If there are some, could authors extract trustable conclusions from them and predict if prebiotics, probiotics and synbiotics are beneficial or not for depression?
Author Response
Reviewer 1
- Authors should emphasize the novelty of their study.
This is an updated review that includes both observational studies and interventional trials published from January 1, 2016 to January 6, 2022. We have included an additional 13 interventional trials and 16 observational studies published since the last review of both observational and interventional studies conducted by Sanada and colleagues (2020) (PMID: 32056863). The gut microbiome is a rapidly changing field with changing conclusions, and our review synthesizes these new findings, providing stronger evidence for the potential benefit of prebiotic, probiotic and synbiotic supplements.
Our article is also one of the few reviews that include prebiotics, probiotics and synbiotics. The most recent synthesis that considered each of these supplements was conducted by Hofmeister and colleagues in 2021 (PMID: 34933877). However, it included patients with comorbid medical and psychiatric disorders who may have differing gut microbiome compositions than patients with depression and healthy controls, and did not review evidence from observational studies.
Our analysis therefore offers a more holistic and up-to-date assessment of the evidence than previous syntheses.
We have updated Section 1.3 to emphasize the novelty of this work:
“Several reviews have also demonstrated a relationship between the GMB and major depressive disorder in human participants [26,42–50]. However, these studies tend to be restricted to patients who meet strict criteria for MDD, at the expense of including more common, subclinical forms of depression that are prevalent in inpatient and healthy populations [51,52] and may be more likely to respond to prebiotic, probiotic and syn-biotic supplementation [53]. In a recent systematic review and meta-analysis, Hofmeister and colleagues (2021) did report a statistically significant benefit of probiotic, prebiotic, and synbiotic interventions in people experiencing depressive symptoms (irrespective of MDD diagnosis) [48]. It is important to note, however, that they included patients with comorbid medical and psychiatric disorders who may have differing gut microbiome compositions than patients with depression and HC, and did not review evidence from observational studies [48].
The most recent systematic review of both observational studies and interventional trials was conducted by Sanada and colleagues in 2020 which included studies published until October 2019 [43]. The review synthesized evidence from ten observational studies that investigated differences in GMB diversity and taxa abundance in patients with MDD compared to HC, and six clinical trials that investigated changes in depressive symptom severity following probiotic or synbiotic administration. The authors report an overall effect of prebiotic and probiotic treatment on depressive symptoms, but inconsistent findings on GMB differences between MDD patients and healthy controls at the phylum level. Since then, an additional 13 interventional trials and 16 observational studies have been conducted which provide new evidence for our analysis. Thus, the present sys-tematic review seeks to understand how the microbiota composition of patients with MDD or depressive symptoms differs from healthy controls, and the potential effects of prebiotic, probiotic and/or synbiotic treatment on depressive symptoms using an updated body of literature.”
- Line 304.
We have now stated the number of studies that report this finding:
“Five of the interventional trials also included microbiota analysis [27,80,83,88,89]. The majority of these studies (four out of five) reported no significant differences in alpha and beta diversity following probiotic administration.”
- Line 340 and Line 347 - study by Nikolova et al., 2021.
We have removed the discussion on the study by Nikolova and colleagues (2021) and contextualized the results using another synthesis (McGuinness et al., 2022; PMID: 35194166), which also reported no significant differences in alpha diversity.
“Our findings are consistent with those of a recent systematic review of gut micro-biome composition in patients with MDD, BD and SZ compared to HC. The authors observed no strong evidence for a difference in the alpha diversity of bacteria in those with any of the above psychiatric disorders compared to controls [91]. Previously, a de-crease in alpha diversity was hypothesized to exist in patients with psychiatric disorders. This was in line with the assumption that greater species number and diversity con-tributed to metabolic functional redundancy and resistance to pathogenic colonization [92,93], preventing disease. More recently, however, human gut microbiome studies suggest that there is limited utility of alpha diversity metrics in measuring gut health and distinguishing disease cases and controls. Equivocal alpha diversity findings have also been observed in neuropsychiatric diseases with similar pathophysiologies, including Parkinson’s disease [94], autism spectrum disorder [95] and anxiety [96].”
- Line 417 - Explain why the evidence is mixed.
We have expanded our discussion to explain the observation of mixed findings:
“One possible explanation for these mixed findings, is that the majority of inter-ventional trials published to date include patients who meet strict criteria for MDD. It is possible, however, that patients with mild depression may derive more benefit from probiotic and synbiotic treatment than those with chronic, treatment-resistant depression [53]. Additional studies in subsamples of patients with depression would be helpful in elucidating the benefits of these treatments.”
We also comment on study design (e.g. placebo control vs. antidepressants), patient sex, and trial length as potential sources of bias in the limitations section. We have added the following paragraph to the discussion to further explain mixed results:
“An additional limitation of this synthesis is that we analyzed the overall effect of various probiotics, prebiotics, and synbiotics. Ideally, multiple studies for each specific compound should be performed and analyzed separately. These studies would discern whether specific compounds influence depressive symptoms while others do not. It would also be valuable to understand the pharmacokinetics, pharmacodynamics, and mecha-nism of action of these supplements. For example, do the probiotics engraft in the gut (pharmacokinetics) and lead to a change in microbial composition and metabolites (pharmacodynamics) with a direct effect on mood? Additional studies are needed to determine these effects.”
- Use conclusions from better designed studies to predict the efficacy of probiotics.
We added a discussion on positive findings from a study on treatment-naive patients to substantiate the first paragraph of section 4.3. It supports the idea that concurrent antidepressant use may affect response to prebiotic, probiotic and synbiotic supplementation.
“Moreover, while antidepressant medications, including tricyclic antidepressants (TCAs), monoamine oxidase inhibitors (MAOIs), and selective serotonin reuptake inhibitors (SSRIs), have been found to have antimicrobial effects, contributing to GMB changes, including dysbiosis [115,116], few studies included drug-naïve patients during the time of the intervention. Indeed, one eight-week pilot trial of a probiotic in treatment-naïve patients with depression (which was included in this review) was associated with sig-nificant improvements in affective clinical symptoms at four and eight weeks follow-up [53].”
Reviewer 2 Report
In the present study findings from observational studies provide evidence for a specific gut microbial profile of patients with major depression disorder as well as the effect of Prebiotics, probiotics, and synbiotics in these patients.
My comments are bellow:
Title: seen results and conclusion would change to "potential benefits of probiotics".
In my opinion, the introduction is too short and does not give enough background on the subject.
In table 5
- Put in italics the name of the species following the taxonomy rules.
- Indicate the number of subjects (n) used in the study and the sex of the participants.
Line 342: Alpha diversity refers to species richness, not only to bacterial diversity.
Table 1: Define MDD
Prebiotics, probiotics, and synbiotics are different things, they should be in separate tables and discussed independently.
Author Response
Reviewer 2
- Change title to “potential benefits of probiotics”.
Thank you for this suggestion. As this paper is focused on prebiotic, probiotic and synbiotic treatment, we believe that the current title better captures the scope of the work.
- The introduction does not provide enough background information on the subject.
We incorporated an additional paragraph in section 1.1 defining dysbiosis, describing a few of the mechanisms by which gut microbiome changes can affect the brain, and suggesting a review article for further reading.
“During dysbiosis, or a disruption to microbiota homeostasis, gut-brain pathways are dysregulated and associated with neuroinflammation and altered permeability of the blood-brain barrier [17]. Microbiota alterations may produce changes in depression by directly affecting release of the neurotransmitters serotonin and dopamine, influencing the stress response and hypothalamus-pituitary-adrenal (HPA) axis, influencing levels of brain-derived neurotrophic factor (BDNF) and triggering the release of inflammatory cytokines [18]. For example, depression is associated with the release of C-reactive protein (CRP) and cytokines such as IL-1, IL-2, IL-6, IFN-γ, and IL-1β [19]. For an in-depth review of the gut brain axis, see this 2019 review by Cryan and colleagues [20].”
- Table 5 - italicize species and indicate N and sex.
As suggested, we have italicized the species in Column 3. We have added the sample sizes (N) in Column 2 (“Population”). We did not include sex as this demographic variable has already been summarized in Table 4.
- Line 342 - Alpha diversity refers to species richness, not only to bacterial diversity.
We have updated the description in the text from “bacterial richness” to “species richness”.
“As alpha diversity is a measure of species richness and evenness within a single population [55], these findings suggest that the diversity of the GMB is similar for patients with MDD and HC.”
- Table 1 - Define MDD.
We have defined the acronym MDD.
- Prebiotics, probiotics and synbiotics should be in separate tables and discussed independently.
We agree that there are important differences between prebiotics, probiotics and synbiotics and their effect on the gut microbiome. In the interest of reducing the number of tables in the manuscript we combined the findings from prebiotics, probiotics and synbiotics into Tables 4 and 5. However, as per the reviewer’s recommendation, we have discussed the prebiotic, probiotic and synbiotic studies independently in the results section, as well as the discussion.
Results:
“More than half (10 out of 17) of the probiotic studies demonstrated a significant decrease in the depressive symptoms of patients treated with probiotics over time, while six reported no significant decrease. In addition, one study reported mixed results, where there was a significant decrease in MADRS score in the probiotic group between baseline and four weeks, but no significant decrease between four and eight weeks [53]. Of the three studies examining the benefits of prebiotic treatment, none reported significant decreases in depressive symptom scores over an eight week follow-up period [78,81,82]. However, significant decreases in symptoms were observed following synbiotic treatment for eight weeks in the single synbiotic study by Ghorbani and colleagues [76].”
Discussion:
“When analyzed together, the interventional trials show a modest benefit of probiotic and synbiotic, but not prebiotic treatment in reducing depressive symptoms of patients with MDD over four to nine weeks relative to placebo. Ten probiotic studies with a combined sample size of 543 participants, five of which were double-blind RCTs and four of which were open-label trials, demonstrated a significant decrease in depressive symptoms over time relative to placebo or antidepressant medication. However, seven high quality studies (N = 462), which were either double-blind or triple-blind RCTs, demonstrated no significant changes in depressive symptoms over time compared to placebo. None of the prebiotic studies demonstrated significant changes in depressive symptoms following intervention. The evidence therefore supports some benefit of probiotic, and synbiotic treatment in patients with MDD relative to placebo, but is largely equivocal.”
Reviewer 3 Report
The present review paper presents important and thorough results regarding the effect of probiotics, prebiotics, and synbiotics administration in patients with depression. The manuscript presents valuable information but it also has to be improved.
- Although the abstract is very well and comprehensively written, it has to be shortened according to journal guidelines: "The abstract should be a total of about 200 words maximum." - Currently it has more than 300 words.
- line 77 - MDDand - please correct
- This section needs references: https://doi.org/10.3390/ijerph19031208
https://doi.org/10.3389/fimmu.2020.604179 - Please revise the whole manuscript regarding the use of italics (i.e. line 254, 294)
- Tables 1, 2 & 3 should be reorganized and I suggest moving the study column at the end - additionally, the references should be inserted based on journal guidelines: "References must be numbered in order of appearance in the text (including table captions and figure legends) and listed individually at the end of the manuscript."
- Also, the tables could be arranged in landscape format, for better understanding
- line 295 - correct FamilactH
- line 346 - Please define the abbreviations. Although IBS was defined in one table, IBD does not appear anywhere. Every abbreviation has to be defined at first use.
Recent studies that also analyzed the same supplement addition in the case of IBS and IBD might present interesting additions:
https://doi.org/10.3390/nu13062112
https://www.eurekaselect.com/article/105166 - line 414 - the abbreviation BDI was defined in the table, but not in the text. These terms have to be defined. It can be also confusing as Body Mass Index is also abbreviated BDI.
Author Response
Reviewer 3
- Shorten the abstract to about 200 words.
We have shortened the abstract from 325 to 194 words.
- Line 77 - Correct MMDand.
We have made the correction.
“While human interventional trials are only beginning to gain traction, preliminary evidence from observational studies has shown that the gut microbiome (GMB) profiles of patients with MDD and depressive symptoms differ significantly from healthy controls (HC) [23–27], as well as patients with other mood [28,29] and anxiety disorders [30].”
- References needed (https://doi.org/10.3390/ijerph19031208 and https://www.frontiersin.org/articles/10.3389/fimmu.2020.604179/full).
We have incorporated the first article into section 1.2 of the manuscript.
“The scientific literature classifies prebiotics as functional foods, given their role in promoting health and preventing disease [33].”
We have incorporated the second article into section 1.1 of the manuscript:
“During dysbiosis, or a disruption to microbiota homeostasis, these pathways are dysregulated and associated with neuroinflammation and altered permeability of the blood-brain barrier [17].”
- Review the manuscript for use of italics.
We have reviewed the entire manuscript for proper use of italics, in accordance with the guideline that when writing scientific names, family, genus, species, and variety or subspecies are italicized. Kingdom, phylum, class, order, and suborder are not italicized.
- Reorganizing Tables 1, 2, 3.
We have retained the study column as the first column of the table to preserve the readability of the manuscript, but inserted the references in journal format in the last column according to the journal formatting guidelines. The page orientation for the tables is already landscape. We have moved the existing Table 3 to the Supplementary Materials section and renamed it Table S1, and added a new Table 3 which summarizes the main taxa abundance findings from the observational studies.
- Line 295 - Correct FamilactH.
We have made the correction.
“Only one study [76] compared the benefits of a synbiotic (Familact H, a combination of fructooligosaccharide and the species L. casaei, L. acidofilus, L. bulgarigus, L. rhamnosus, B. breve, B. longum, and S. thermophilus) relative to placebo.”
- Line 346 - Define the abbreviations IBS and IBD.
We have now defined the abbreviation IBS.
“For both observational studies and clinical trials, we excluded (1) studies published before 2016, (2) reviews, case reports, conference abstracts, dissertations, or letters, (3) protocol descriptions of studies not yet conducted, and (4) studies reporting only pooled results in patients with comorbid psychiatric and medical conditions (e.g. bipolar disorder (BD), schizophrenia (SZ), anxiety, and irritable bowel syndrome [IBS]).”
Due to other revisions, IBD is no longer mentioned in this manuscript.
- Additional articles (https://www.mdpi.com/2072-6643/13/6/2112 and https://pubmed.ncbi.nlm.nih.gov/32164516/).
Thank you for this suggestion. We made changes to this section of the manuscript based on another reviewer’s recommendation, and consequently the discussion on IBS and IBD was removed. We therefore did not incorporate these articles into the revised manuscript.
- Line 414 - Define the abbreviation BDI.
We have now defined this abbreviation in the text.
“Depressive symptoms were most often assessed by the Beck Depression Inventory (BDI) and HAM-D.”
Round 2
Reviewer 1 Report
Regarding the response, Line 340 and Line 347 - study by Nikolova et al., 2021. “We have removed the discussion on the study by Nikolova and colleagues (2021) and contextualized the results using another synthesis (McGuinness et al., 2022; PMID: 35194166), which also reported no significant differences in alpha diversity.”
The Nikolova’s study should be also included in the Discussion section. Authors should compile the studies that agree and differ from their findings.
Author Response
Thank you for taking the time to re-review this manuscript. As per your suggestion we have included the study by Nikolova and colleagues (2021) in the discussion:
“Our findings are consistent with those of a recent systematic review of gut micro-biome composition in patients with MDD, BD and SZ compared to HC conducted by McGuinness and colleagues (2022) [91]. The authors observed no strong evidence for a difference in the alpha diversity of bacteria in patients with psychiatric disorders com-pared to HC [91]. Previously, a decrease in alpha diversity was hypothesized to exist in patients with psychiatric disorders. This was in line with the assumption that greater species number and diversity contributed to metabolic functional redundancy and re-sistance to pathogenic colonization [92,93], preventing disease. More recently, however, human gut microbiome studies suggest that there is limited utility of alpha diversity metrics in measuring gut health and distinguishing disease cases and controls. The evidence for alpha diversity changes in patients with MDD relative to HC is in fact largely mixed. For example, while McGuinness and colleagues (2022) reported no significant differences in alpha diversity between patients with psychiatric illness and HC [91], in a recent systematic review and meta-analysis of a pooled sample of patients with MDD and comorbid mental illnesses, Nikolova and colleagues (2021) observed significant differ-ences in alpha diversity relative to HC [94]. Equivocal alpha diversity findings have also been reported in neuropsychiatric diseases with similar pathophysiologies to depression, including Parkinson’s disease [95], autism spectrum disorder [96] and anxiety [97].”
Reviewer 3 Report
The article was considerably improved and reorganized.
In tables what I still suggest is to merge the first and last columns (Study and reference). For example: Aizawa 2016 [58].
Otherwise, the article can be published it is well written and of major importance.
Author Response
Thank you for taking the time to re-review this manuscript. As suggested, we have merged the first and last columns of Tables 1, 2, 4, and 5.